# Rice Husk-Based Insulators: Manufacturing Process and Thermal Potential Assessment

**DOI:** 10.3390/ma17112589

**Published:** 2024-05-28

**Authors:** Luis Cigarruista Solís, Miguel Chen Austin, Euclides Deago, Guillermo López, Nacari Marin-Calvo

**Affiliations:** 1Department of Mechanical Engineering, Universidad Tecnológica de Panamá, Panama City 0819-07289, Panamamiguel.chen@utp.ac.pa (M.C.A.); guillermo.lopez2@utp.ac.pa (G.L.); 2Research Group—Iniciativa de Integración de Tecnologías para el Desarrollo de Soluciones Ingenieriles (I2TEDSI), Panama City 0819-07289, Panama; 3Centro de Estudios Multidisciplinarios en Ciencias, Ingeniería y Tecnología (CEMCIT-AIP), Panama City 0819-07289, Panama; 4Sistema Nacional de Investigación (SNI), Clayton 0816-02852, Panama; euclides.deago@utp.ac.pa; 5Centro de Investigaciones Hidráulicas e Hidrotécnicas (CIHH), Universidad Tecnológica de Panamá, Panama City 0819-07289, Panama; 6Research Group—Biosólidos Energía y Sostenibilidad (BioES), Panama City 0819-07289, Panama

**Keywords:** building sustainability, natural fibers, resource sustainability, rice husk waste, roof insultation, thermal insulation

## Abstract

The development of bio-insultation materials has attracted increasing attention in building energy-saving fields. In tropical and hot–humid climates, building envelope insulation is important for an energy efficient and comfortable indoor environment. In this study, several experiments were carried out on a bio-insulation material, which was prepared by using rice husk as a raw material. Square rice husk-based insultation panels were developed, considering the ASTM C-177 dimensions, to perform thermal conductivity coefficient tests. The thermal conductivity coefficient obtained was 0.073 W/(m K), which is in the range of conventional thermal insulators. In a second phase of this study, two experimental enclosures (chambers) were constructed, one with rice husk-based insulation panels and the second one without this insulation. The measures of the temperatures and thermal flows through the chambers were obtained with an electronic module based on the ARDUINO platform. This module consisted of three DS18B20 temperature sensors and four Peltier plates. Daily temperature and heat flux data were collected for the two chambers during the dry season in Panama, specifically between April and May. In the experimental chamber that did not have rice husk panel insulation on the roof, a flow of up to 28.18 W/m^2^ was observed, while in the chamber that did have rice husk panels, the presence of a flow toward the interior was rarely observed. The rice husk-based insulation panels showed comparable performance with conventional insulators, as a sustainable solution that takes advantage of a local resource to improve thermal comfort and the reduction of the environmental impact.

## 1. Introduction

The constant increase in global warming has led to a rise in temperatures, prompting people to develop strategies to mitigate the resulting impacts. By implementing insulation in homes, benefits such as improved thermal comfort, reduced heat losses, enhanced energy efficiency, and reduced air-conditioning costs can be achieved. Heat gain in an edification is primarily due to heat transfer through the building envelope. Windows, walls, and roofs are the areas most prone to have heat transfer with the environment. A significant amount of energy in the form of heat enters a space through the roof, as it is the part where the most energy is concentrated by radiation and convection due to continuous solar exposure during the day [1,2]. Critical heat gains through the roof were obtained when performing yearly monthly averaged simulation in Panama’s climate conditions (between 780 and 900 kWh) [1]. Therefore, it is important to conduct a study that allows for proposing solutions related to this issue.

Natural vegetable fibers have shown excellent results in improving living conditions within a residence [3,4,5,6,7,8,9,10,11]. Natural fibers have been used as building materials, such as high-strength concrete [12,13,14,15], seismic reinforcements [16], and thermal insulation [17,18,19].

Rice husk has been the subject of study due to its thermal insulation properties [3,4,6,7,8,15,17,20,21,22,23,24,25,26,27,28,29]. The studies conducted have demonstrated the good insulating properties when mixed with different types of binders, such as paper pulp cellulose, where a thermal conductivity of 1.08 W/m K was obtained. Similarly, natural fibers mixed with corn starch as a binder present a thermal conductivity of 0.0262 W/(m K) [30]. Rice husk is the result of agricultural waste processes, which have a significant production in Panama. About 98,040 hectares of rice crop are cultivated in Panama and the rice husk harvest reached 812 740,000 kg between 2020 and 2021 [31]. Approximately 20% of the rice paddy is rice husk, sometimes discarded by burning or disposed of in municipal landfills annually, causing waste accumulation and environmental deterioration [23,32,33].

The role of thermal insultation materials in the building envelope is significant, especially in hot–humid regions [17], and Panama is no exception. The need for climatization indoors represents between 30 and 40% of energy consumption [10,34,35]. This highlights the need to develop innovative and sustainable solutions that take advantage of local resources.

The application of green roofs as an adaptation and reduction strategy concerning the effects of climate change is increasingly attractive in the research and building sectors. Several layers of materials are implemented as a strategy that contributes to the thermal insulation of the roof and therefore improves the thermal insulation of a building. In Honduras, different layers that included PEAD plastic, expanded polystyrene, saran mesh, substrate, and vegetal cover were implemented in small-scale rooms to which this insulation method was applied [36]. The temperature of each room with and without roof insulation was measured for 19 days, which were distributed over a period of 5 weeks. The data collection period was from 7:00 am to 7:00 pm, to record during daytime and nighttime periods, maintaining a 15 min interval between each reading. Among the results obtained, it can be mentioned that the ambient temperature directly influences the internal temperature of the sample units. Also, heat transfer on green roofs presents a reduction of up to 4.22 °C [36].

An investigation into the use of straw from different kinds of cereals in the form of insulating or structural panels is presented in [37]. A chamber was constructed using this natural fiber. According to this work, the standard insulation capacity of straw presents double the insulation required by some European standards, with a reduced energy consumption of up to 50%. The thermal conductivity obtained from the developed material varied between 0.07 and 0.09 W/(m K) [37].

Thermal analysis of two rural roofs from Amazonian homes, one with an Irapay leaf roof and the other one with a galvanized steel sheet roof, were presented in [38]. An analysis was conducted during the month of May at the hottest hours of the day. Temperature tests were carried out on the roof surfaces, between 12:00 pm and 2:00 pm, over a period of 10 days. Temperature measurements of the bottom parts of the roofs were taken at 50 min intervals. As a result, it was concluded that the room with a roof made from Irapay leaves had a lower interior temperature, contributing to reducing the thermal stress and improving the comfort within the room [38].

A study on the variation in the thermal conductivity of rice husks when agglomerated with vegetable fibers is presented. Various fibers were used to create different compositions at different proportions. The fibers used were rice husk, flax fiber, banana fiber, and yuca fiber. For the measurement of thermal conductivity, the ASTM C-177 standard [39] was used. Each fiber was separately exposed to a heat flow until it reached a steady state. The results obtained were positive in terms of the thermal insulation properties, with the best result being the mixture of rice husk, yuca starch, banana fiber, and flax fiber (0.0653 W/(m K)) [40]. In another study, rice husk ceiling boards and potato starch in combination with Arabic gum as adhesives presented acceptable values and were within the established range for roof insulation (0.023–2.9 W/(m K)) [27]. A study focused on the thermal analysis of granulated rice husk presented a three step procedure: collection, washing, and drying of the raw material. The sample with the highest percentage of silica in its chemical composition yielded the lowest thermal conductivity value (0.0746 W/(m K)). This indicates that silica is a fundamental component for the thermal properties of granulated rice husk [23].

This study aims to produce an insulating panel using rice husk with a commercial rice flour as a binder. It is analyzed in terms of its potential as a thermal insulator when applied to galvanized steel roofs of residential buildings. This study includes the manufacture of panels, followed by the installation of these panels in a chamber to evaluate their thermal insulation capacity. The panels were placed under a steel sheet roof (called “zinc”, because of the protective galvanized layer), a material commonly used for residential and industrial roofs in Panama.

Two experimental chambers were constructed: the first one served as a reference, where no insulating material was installed under the steel roof, and in the second one, the rice husk panels were installed. Temperature readings were taken inside and outside each chamber. Thermal flow measurements were also taken on the upper and lower surfaces of the roofs, during the period between April and May, months considered the hottest of the year on the Azuero Peninsula, Panama.

This research aims to promote the use of materials considered to be agro-industrial waste in Panama, demonstrating their applicability in a different area and contributing to saving energy and improving comfort indoors.

## 2. Materials and Methodology

### 2.1. Methodology for Rice Husk Panel Manufacture

The following materials were used to prepare the different specimens of ground rice husk (supplied by Cooperativa de Ahorro y Crédito El Avance. R. L., La Villa de Los Santos, Los Santos, Panama) and commercial rice flour (Crema de Arroz, Molino, Panama City, Panamá) as the binder. All the raw materials were obtained in Chitré, Azuero Peninsula, Panamá. The methodology for the rice husk panel manufacture is based on previous experiences of the authors [17,41,42]. To summarize, the steps taken to obtain a rice husk panel were as follows: The rice husk was ground, and the resulting material was weighed. The binder was three parts water and one part commercial rice flour, cooked together at medium heat until it reached a cream consistency. The grounded rice husk was mixed with the prepared binder (for every two parts ground rice husk three parts binder was added). This mixture was combined using a mortar mixer until a uniform distribution of the components was obtained. Once the mixture was obtained, it was transferred to a mold. An aluminum mold with the desired dimensions (30 cm wide, 30 cm high and 1.5 cm thickness) was used, according to the ASTM C-177 thermal conductivity coefficient test [39,43]. A metal mesh was added to avoid the presence of air inside the mixture and the panels’ deformation during the drying process. The drying process was performed in an artisanal oven, preheated to 220 °C for 10 min. Then, the rice husk panel mixture was prepared inside the oven for 3 h. The mold was removed from the oven and cooled at room temperature. After the cooling process, the rice husk panel was demolded and prepared for the thermal conductivity test. Figure 1 shows the binder consistency of the commercial rice fluor (a), the aluminum mold and rigid metal mesh used during the manufacturing process (b) and the obtained rice husk panel (c).

### 2.2. Measurement of the Thermal Conductivity of the Rice Husk Panel

To measure the thermal conductivity of the obtained panel, the experiment was conducted using the ASTM C-177 standard [39,43]. This standard is based on Fourier’s heat conduction equation:(1)$$\dot{Q}=-KA\frac{dT}{dx}$$
where $\dot{Q}$ represents the amount of heat transferred through the material in Watts, *A* is the area in m^2^, *K* is the material’s thermal conductivity coefficient in W/(m K), and dT/dx is the temperature gradient in the direction of the heat flow. To achieve Equation (1), a steady-state thermal condition must be reached. In this experiment, the aim was to obtain the value of the thermal conductivity coefficient (*K* value), which means all the other variables in the equation must be constant over time. The experiment consisted of a fully isolated wooden box on the inside, insulated by an aluminum-coated paper insulation layer. This box, referred to as the “hot box”, was designed to store the heat generated inside until it reached a steady-state thermal condition. The heat was generated by a 100 W incandescent bulb installed perpendicular to one of the adjacent faces of the box. The manufacturer of the incandescent bulb stated in its technical specifications that only 85% of the specified power was converted into heat, while the rest was converted into light energy. The dimensions of the “hot box” were 30 cm wide, 30 cm high, and 30 cm deep. Figure 2 shows the “hot box” (a), the placement of the test material in the “hot box”, along with the computer used to obtain the temperature data from the sensor (b) and a thermographic image obtained from the tests (c).

To determine the interior temperature of the “hot box”, a DHT22 temperature sensor (Aosong Electronics Co., Ltd., Guangzhou, China) was placed inside it. It was programmed using an ARDUINO microprocessor. The microprocessor was responsible for validating the steady-state thermal condition inside the box. The procedure involved placing the material from which the thermal conductivity was to be determined perpendicular to the incandescent bulb [43]. The material was arranged to have the same dimensions as the box and a thickness of 1.5 cm. The edges of the test material were joined to the box using silicone and adhered with 3 M tape for better attachment.

The DHT22 sensor has the capability to simultaneously record temperature and humidity data. This sensor operates at a maximum current of 2.5 mA and handles a temperature range from −40 to 80 °C. The temperature accuracy is <+ −0.5 °C [44]. Due to the thermal conductivity experiment involving high temperatures within the “hot box”, a temperature reading correction factor was used for experiments exceeding the manufacturer’s indicated temperature limits. This correction factor was determined through the measurement of the ambient temperature and relative humidity at the test location where the experiment was performed. To obtain the correction factor, the ambient temperature at the test location was measured and divided by the constant equivalent to that temperature, considering the ambient temperature of 30 °C and the corresponding resistance value for the indoor and outdoor ambient temperature sensors (12.07). Dividing the temperature by the resistance value results in the correction factor of 2.4855. The temperature provided by the sensor was automatically multiplied by this factor, thus providing the actual temperature inside the “hot box”. The previous procedure was established, performing tests with different materials with known *K*-value, under the same ambient conditions [43].

As the experiment began, it was observed that as the box’s temperature increased and the rice husk panel exhibited a slight bow of approximately 0.5 cm in its central part. For this reason, the central part was secured with a piece of string to prevent any minimal heat leakage. The temperature readings were maintained for 1 h and 30 min with a 1 min interval between each. Over that time, the temperature increased until it reached 230 °C inside the “hot box” and remained constant for a period of 10 min. At that point, it could be assumed that the hot box had reached a steady-state thermal condition inside. As observed in Equation (1), the gradient of the surface temperature of the test material needed to be obtained. Due to the steady-state thermal condition inside the box, the value of the surface temperature on one of the material surfaces was known. To obtain the temperature of the external surface, a FLUKE Ti 110 infrared thermometer (Hebi, China) was used [17,42,43].

The color difference in the image is due to the direction of the heat flow inside the box. The reddish part is due to the heat flow generated by the incandescent bulb being perpendicular to that part of the panel. The external temperature of the rice husk panel was 36.3 °C. With the previously mentioned data, all the necessary variables were available to satisfy Equation (1). The next step was to solve *K* in Equation (2).
(2)$$K=\frac{L}{A(T_{1}-T_{2})}\dot{Q}$$

In Table 1, the data for the variables are presented to obtain the thermal conductivity coefficient. The heat transferred through the material was 85 W, obtained from the specifications of the incandescent bulb used.

Applying Equation (2), a value of *K* = 0.073 W/(m K) was obtained. The obtained thermal conductivity value falls within the range that indicates when a material is a good thermal insulator. Materials considered good thermal insulators have a thermal conductivity value of less than 0.080 W/(m K).

The TERMOFLUX LC module was designed to measure two variables simultaneously. It was built on the ARDUINO digital platform, featuring an Arduino UNO board, 3 DS18B20 temperature sensors (Dallas Semiconductor, acquired by Maxim Integrated, San Jose, CA, USA), 4 PELTIER TEC1-12706 plates, a 4.7 KΩ resistor, and a microSD module for daily data storage (Figure 3). The module was constructed on a circuit board tester, commonly known as a protoboard, and was protected by a wooden case to prevent overheating due to continuous solar exposure during the measurement period. The measurement cycle for the variables involved repeated every 5 min over a 7 h daily period, providing a total of 42 temperature and thermal flow measurements.

### 2.3. Experimental Chambers Design

The materials used for the construction of the experimental chambers were as follows: galvanized steel sheet (zinc), EPS insulation walls to minimize the heat gains, compressed cardboard sheets, cement and sand. Figure 4 shows the design and dimensions of the experimental chambers: 1.20 m × 1.00 m × 1.30 m. Under the galvanized roof, the rice husk panels were placed under the galvanized roof, then the compressed cardboard sheets were installed under the rice husk panels as the insulation system in one of the chambers (Figure 4b). Figure 5 shows this configuration, where the ceiling/compressed cardboard sheets were placed under the insulation material (rice husk panels). These panels were placed under the galvanized steel roof. A space for air was located between the insulation material and the galvanized steel roof, where the Peltier plate was installed.

The chambers were located in Chitré, Panama, with climate characteristics of a tropical Savana (Aw) Koppen climate classification. This region presents an average annual temperature of 27.2 °C, with a maximum of 35.6 °C. It has a maximum annual relative humidity of 100%, with an average of 78%. It has an average annual wind speed of 2.5 m/s and an average annual global radiation of 4.96 kWh/m^2^-day [45].

Due to the way the TERMOFLUX LC temperature and thermal flow measurement module was designed, it was only capable of performing measurements of positive voltages. Because of the Seebeck effect, an electrical answer was generated when thermal flow passes through the Peltier plate [46]. Disturbances were caused by the reverse direction of the heat flow on the external surface of the roof of each chamber. The roofs of the chambers were made of galvanized steel sheet, indicating that it is a good conductor of heat. Due to the heat conduction properties of the roof material, it reached a much higher temperature on its surface than the upper surface temperature of the Peltier plate. The lower surface of the Peltier plate, which was adhered to the galvanized steel roof, heated up faster compared to the upper surface where the external air flowed [46]. It is known that heat transfer occurs from the body of a higher temperature to the body of a lower temperature, and there must be a temperature difference between the upper and lower surfaces of the Peltier plates for thermal flow. The Peltier plate measured the flow in one direction; it was not capable of capturing the thermal flow passing through the roofs of the chambers due to the reverse direction of the heat flow. The top Peltier plates were installed on the lower surface of the galvanized steel roof and on the lower surface of the insulation material (Figure 5).

For the temperature sensors, a design was made to allow them to be placed exactly in the center of each chamber (Figure 6). The TERMOFLUX LC module was in the center of the two chambers. It was placed inside an insulation box to avoid direct exposure to solar radiation and thus protect the electronic components.

The experimental chambers were built in a north–south orientation. Conventional insulation material, which was compressed cardboard, was installed. The size of each insulation sheet was 118 cm × 90 cm. This material is commonly used because it has good rigidity, and its cost is relatively lower than other materials used for the same purpose. After installing the insulation material, the rice husk panels were placed in one of the chambers. Figure 7 shows the result of the installation of the compressed cardboard together with the rice husk insulation panels. Chamber 1 is without rice husk insulation, whereas Chamber 2 is with rice husk insulation panels installed.

On the outside, the first temperature sensor was installed to measure the ambient temperature. It was placed in front of the chambers hanging from a plastic bar to present a greater exposure to the ambient conditions surrounding the chambers. Inside each experimental chamber, temperature sensors and Peltier plates were installed. They were adhered with thermal paste to the underside of the insulation material. To place the temperature sensors, metal rods were installed to ensure their positioning in the center of the chamber (Figure 8) to ensure that the temperature measured by the sensors inside the chambers was the same temperature as the air inside. If the temperature sensors were placed very close to the insulation material, they could be affected by the heat flow present close to it. In real life, there will be some sort of ventilation, such as windows. The experimental chambers were completely closed to ensure temperature and heat flux measures inside the chambers without the interference of cross flux.

## 3. Results and Discussion

The result of this experiment was a rice husk panel made from ground rice husk and a binder based on commercial rice flour. Small cracks were observed, which were insignificant in comparison to previous results. These were the cause of the compression given to it with the metal grid. Its consistency was entirely solid and compact. It is important to mention that this process was performed in an artisanal oven. The manufacturing process for the rice husk insulation panels can be optimized, considering the size of the ground rice husk particles (with dimensions less than 0.6 mm) and the drying process, leading to a process with commercialization potential [17].

The daily average temperatures were registered and graphed monthly to have a broad overview of the temperature behavior in the chambers against the ambient temperature. Differences in temperatures between the chambers were observed up to 4 °C. In the chamber that did not contain rice husk panel insulation (Chamber 1), the temperature inside was very similar to the external ambient temperature; however, it reached higher levels than the ambient temperature. This was due to how the chambers were built, for experimental control. Ventilation was restricted by external environmental conditions into the chambers, making it less likely for there to be interaction between the indoor air and the outdoor air. It is worth noting that the data registered in April were affected by a few rainy days. In Figure 9, the behavior of the average temperatures of April can be observed. Notable is the similarity of the ambient temperature and the chamber that did not contain insulation in the roof; however, in the temperatures of the chamber that did contain rice husk panel insulation, a difference between these two temperatures is evident. It is worth noting that the graphs of each temperature variable in almost all the readings maintained a trend as the values rose or fell. This means that the sensors provided consistent readings regarding the scenario where they were located.

In May 2021, data on the temperatures of the chambers were collected during climate conditions of sunny and rainy days. As expected, the charts maintained almost the same behaviors in terms of the similarities between the ambient and the non-insulated chamber temperatures and the differences of the variables previously described with the chamber that did contain insulation. A difference of up to 4 °C was achieved between the two experimental chambers (See Figure 10).

Data on the thermal flows passing through the roof of each chamber were collected. To perform the corresponding analysis, a daily average of the collected data was performed. The results from 2 April to 7 April showed inadequate behavior due to the phenomenon of reverse thermal flow. From 8 April onwards, a behavior more in line with the scenarios where the research project was carried out was maintained. The thermal flows that passed through the upper Peltier plate of the chamber without insulation were much higher compared to the thermal flows that passed through the upper Peltier plate of the chamber with insulation. There were average thermal flows of up to 120 W/m^2^ on this plate. The thermal flows can be observed graphically in Figure 11.

Regarding the thermal flows of the lower Peltier plate of the uninsulated enclosure, a thermal flow of up to 28.18 W/m^2^ was observed. This thermal flow was not constant throughout the day as it varied in intervals from zero to the previously mentioned value. Its maximum value was reached in the hourly intervals from 12:00 pm to 2:45 pm on numerous occasions (See Figure 12).

This significant difference in the thermal flows was due to a heat storage effect in the part of the roof where the air layer was located. In the chamber that did not contain rice husk panel insulation, the thermal flow passed through the galvanized steel roof and then entered inside the chamber more easily. In this case, heat was not stored in the air layer of the roof, causing a relatively low temperature. This generated a relatively higher temperature difference between the lower and upper surfaces of the top Peltier plate, facilitating the transfer of heat and making it easier for the thermal flow to pass through the Peltier plate. In the top Peltier plate of the chamber that contained rice husk insulation, the opposite effect was presented. Due to the insulation fulfilling the function of reducing the entry of the thermal flow through the roof into the chamber, it was stored in the air layer present on the roof. This caused the temperature to rise in that part and the temperature gradient between the upper and lower surfaces of the Peltier plate to be relatively smaller compared to the case of the chamber without insulation.

In this study, fire resistance tests were not considered. In a similar study performed by the authors, the addition of borax provided an effective barrier against fire and exhibited promising behavior in terms of fire resistance [47]. For future developments, the authors are evaluating the addition of other elements, such as borax, under controlled conditions, which could improve fire resistance and inhibition of fungal and yeast growth.

Limitations in the development of this project can include those imposed by the COVID-19 pandemic (during 2021 and 2022) and the available equipment, both in terms of the manufacturing process of the rice husk insulation panels. However, this study promotes the use of materials considered as agro-industrial waste in Panama, demonstrating their potential as an insulation material. The proposed material has the potential for thermal insulation, so, for its future commercialization, the research and development process must continue, providing additional improvements. Also, it is important to mention that this study was an effort in the development of an innovative and sustainable solution, considering the use of local resources.

## 4. Recommendation

For future developments, the authors are evaluating the addition of other elements, such as borax, under controlled conditions, which could improve fire resistance and inhibition of fungal and yeast growth. Also, it is considered to perform durability and permeability testing during the rainy season. In the present case, the behavior was evaluated during dry season, a condition that prevails for most of the year in the Azuero region, Panama.

It is necessary to optimize the manufacturing process, considering the size of the ground rice husk particles and the drying process. This could be achieved with specialized equipment, as well as sieves that allow particles with smaller dimensions with the objective of obtaining a more homogeneous material, leading to a process with commercialization potential.

Future studies need to address these two aspects in order to fully assess the implications of this alternative in real-scale cases. Humidity effects in long-term studies also need to take place due to the high-humidity weather.

## 5. Conclusions

The panels made from rice husk exhibited good thermal insulation properties, showing a thermal conductivity coefficient of 0.073 W/(m K). The rice husk-based insulation panels showed a satisfying performance and can be used in buildings to realize energy savings and show great potential as an alternative to replace the conventional material insulators in current use in tropical countries. On the other hand, the elaboration process of rice husk-based insulation panels must be optimized for future commercialization.

The thermal flow and temperature measurement module can provide results with a high percentage of accuracy due to the types of sensors used in the experiments. The Peltier plates used for measuring the thermal flows proved to be an economical option for this purpose. The experimental chambers that were built could provide a favorable scenario for experimentation with the type of insulation being studied.

In the experimental chamber that did not have rice husk panel insulation on the roof, a flow of up to 28.18 W/m^2^ was observed, while in the chamber that did have rice husk panels as insulation on the roof, the presence of fa low toward the interior was rarely observed.

The internal temperatures of the chambers were the result of the constant heat storage that was present throughout each day of measurement. Due to the maximum restriction of the interaction between the external air and the internal air of the chambers, there was no possibility of renewing the internal air. This led to an increase in the temperature; however, the action of the roof insulation was reflected in a temperature difference of up to 4 °C between the two chambers.

Although proving to have comparable performance to conventional insulator materials, the rice husk-based insulator presents other benefits as it is made from organic materials with low carbon emissions at a low cost. Future studies need to address these two aspects in order to fully assess the implications of this alternative in real-scale cases. The humidity effects in long-term studies also need to take place due to the high-humidity weather. This study was an effort in the development of an innovative and sustainable solution, considering the use of local resources, as an alternative to replace the conventional insulators materials in current use in tropical countries.

## Figures and Tables

**Figure 1 materials-17-02589-f001:**
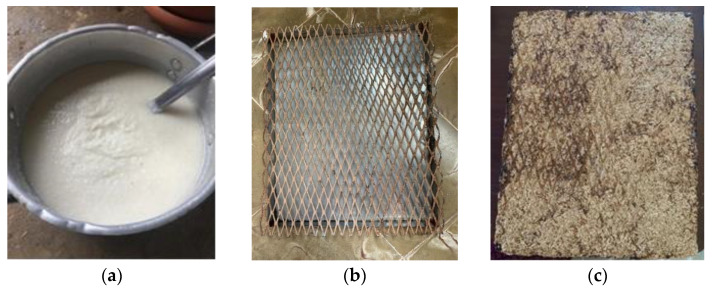
(**a**) Binder consistency, (**b**) aluminum mold and metal mesh used, and (**c**) rice husk panel.

**Figure 2 materials-17-02589-f002:**
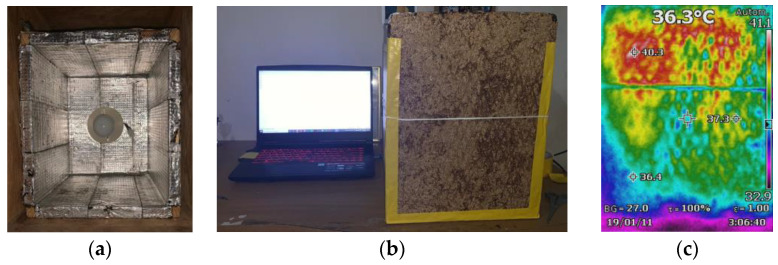
(**a**) Hot box prototype for measuring the thermal conductivity, (**b**) measuring process for the *K* value of the rice husk panel, and (**c**) thermographic image of the rice husk panel.

**Figure 3 materials-17-02589-f003:**
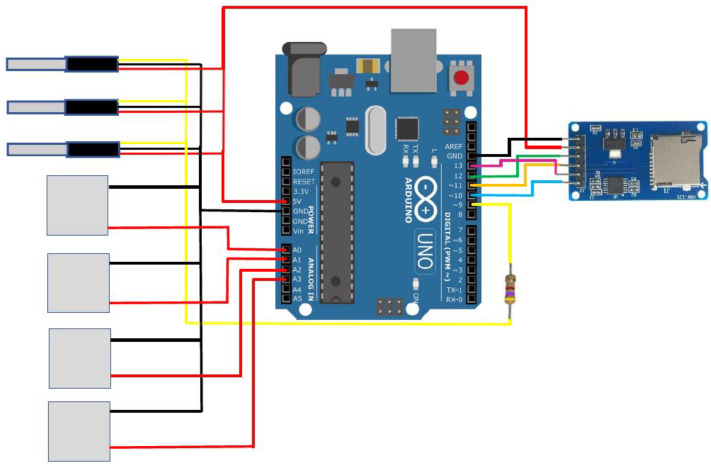
Connection scheme of the TERMOFLUX LC module, DS18B20 temperature sensors and Peltier TEC1-12706.

**Figure 4 materials-17-02589-f004:**
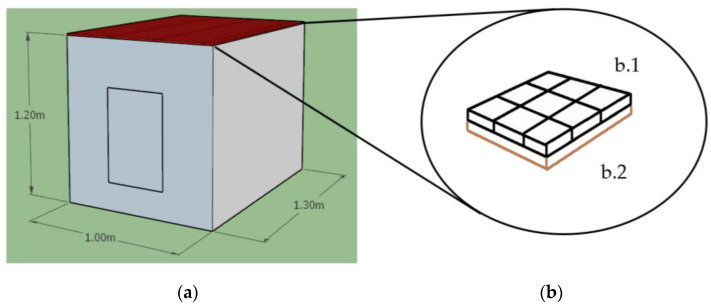
(**a**) Experimental chambers’ dimensions, and (**b**) view under the galvanized roof: (b.1) rice husk panels, (b.2) compressed cardboard (ceiling).

**Figure 5 materials-17-02589-f005:**
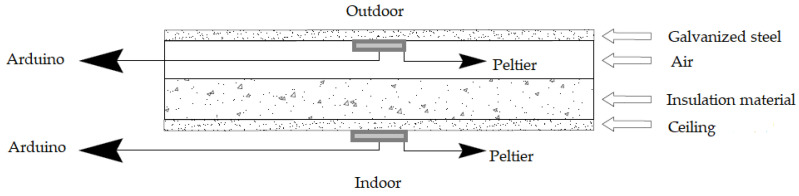
Peltier plates and roof configuration of the insulated chamber.

**Figure 6 materials-17-02589-f006:**
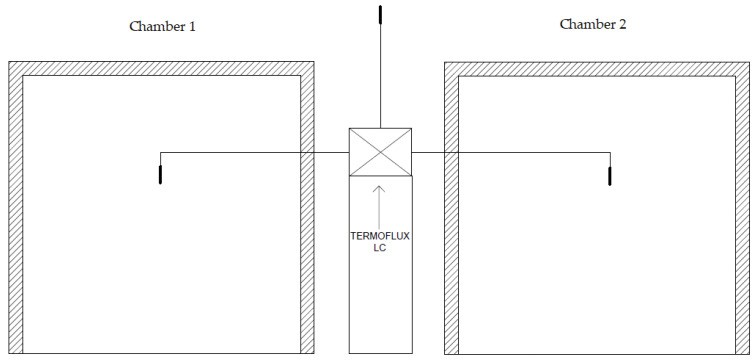
TERMOFLUX LC location in the experimental chambers.

**Figure 7 materials-17-02589-f007:**
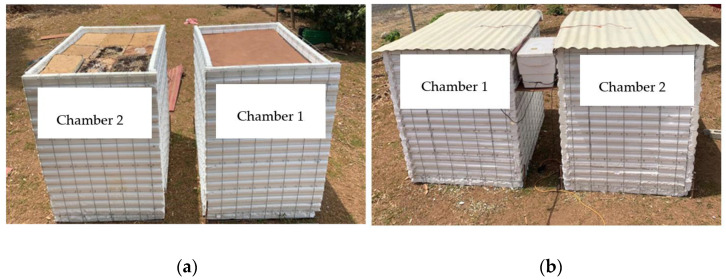
Finished experimental chambers: Chamber 1 without rice husk insulator and Chamber 2 with the rice husk insulator. (**a**) Chambers without galvanized roofs and (**b**) chambers with galvanized roofs.

**Figure 8 materials-17-02589-f008:**
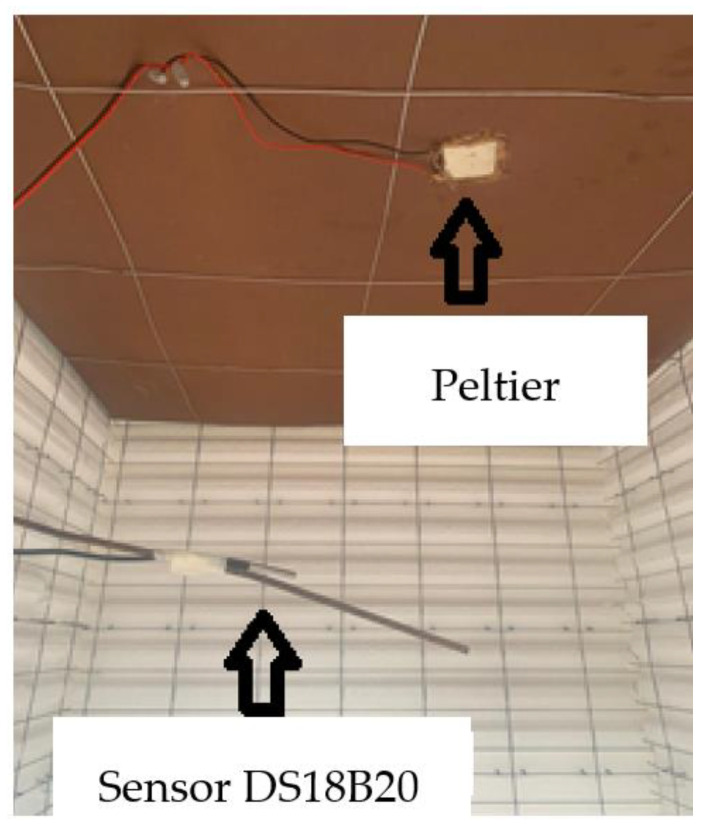
Placement of sensors inside the chambers.

**Figure 9 materials-17-02589-f009:**
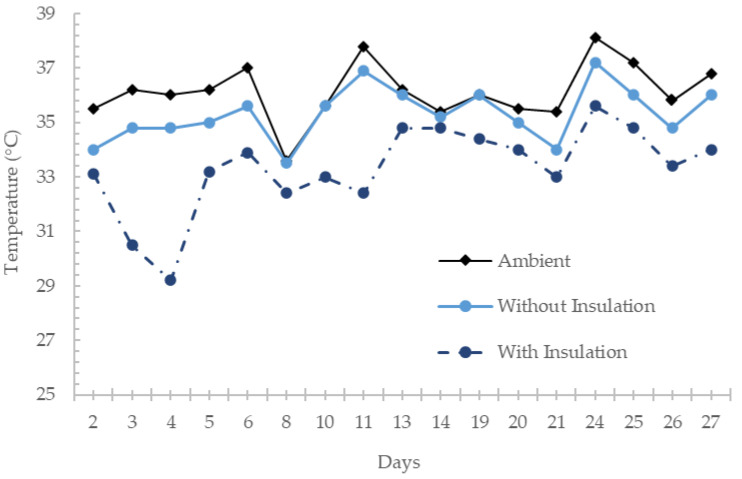
Average temperatures in April 2021.

**Figure 10 materials-17-02589-f010:**
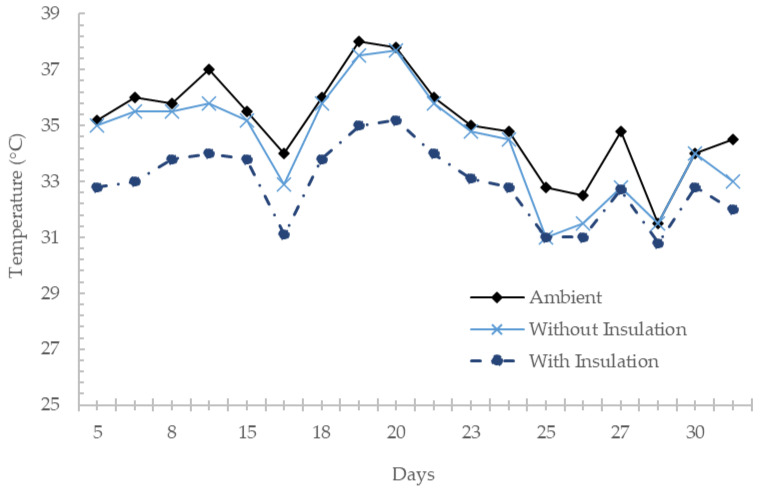
Average temperatures in May 2021.

**Figure 11 materials-17-02589-f011:**
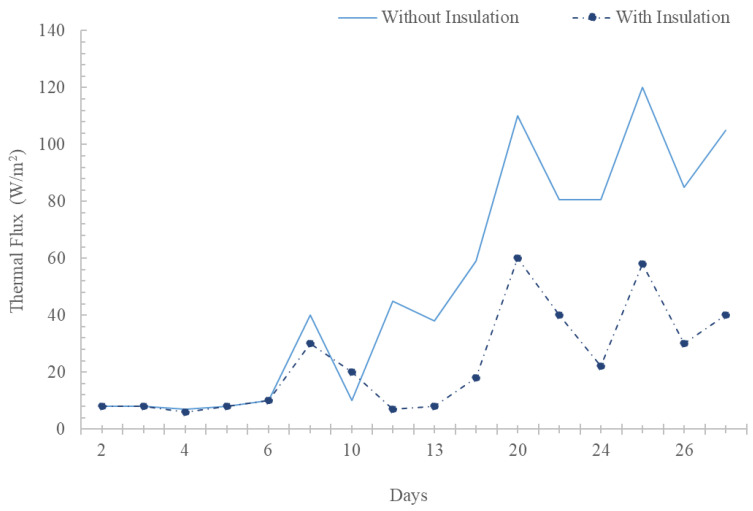
Average thermal flows of the upper Peltier plates for April 2021.

**Figure 12 materials-17-02589-f012:**
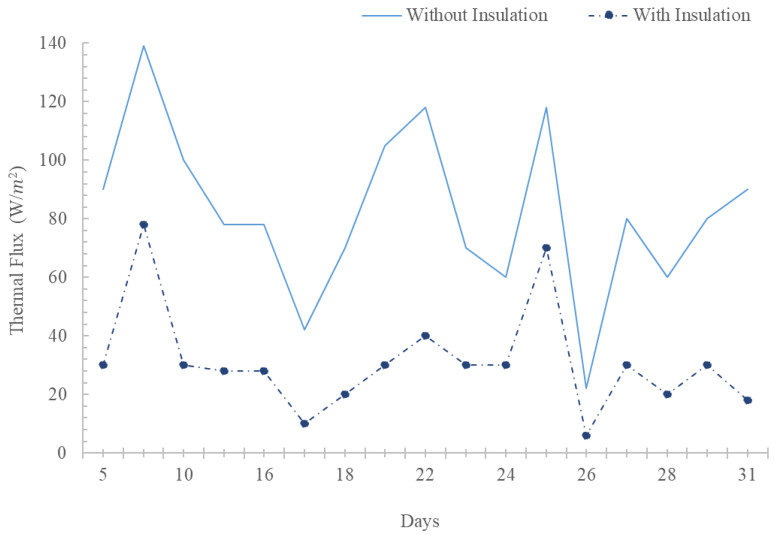
Average thermal flows of the upper Peltier plates for May 2021.

**Table 1 materials-17-02589-t001:** Data for the variables used to obtain the thermal conductivity.

Variable	Value
Heat transferred ($\dot{Q})$	85 W
Thickness (L)	0.015 m
Area (A)	0.09 $m^{2}$
Temperature inside ($T_{1}$)	230 °C (503.15 K)
Temperature outside ($T_{2}$)	36.3 °C (309.45 K)

## Data Availability

The original contributions presented in the study are included in the article, further inquiries can be directed to the corresponding author.

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
