# Peer review of "Rice Husk-Based Insulators: Manufacturing Process and Thermal Potential Assessment"

_materials, 2024, doi:10.3390/ma17112589_

Round 1
Reviewer 1 Report
Comments and Suggestions for Authors
please see attached file.

Minor grammatical mistakes (singular, plural, etc.) and typos (incorrect/missing punctuation, sentences attached to each other, etc.) need to be corrected
Author Response
May 7, 2024
Dear Editors:
I would like to submit the manuscript «materials-2999973 », entitled « Rice husk-based insulators: Manufacturing process and thermal potential assessment » by author names to be reconsidered for publicaction in Development and Characterization of Bio-Based Insulation Materials, as an special issue of Materials (ISSN 1996-1944), section Construction and Building Materials
All the changes are in red, answering to reviewer’s recommendations. The changes based on reviewer recommendations are:
- The tile of the paper is long. I suggest removing “Manufacturing process and thermal
potential assessment”. As anyway, the manufacturing process is very briefly described. In this case, the authors suggest the new title: « Rice husk-based insulators: Manufacturing process and thermal potential assessment»
- Based on the authors work, how these insulation panels can be optimized? Please see lines 402-406 in the revised manuscript.
- Line 59: please replace the word “Environ”. Please see line 62 in the revised manuscript.
- Line 138: why the metal mesh was added?. please rephrase this sentence as it is not
clear. Please see lines 144-148 and figure 1(b) added in the revised manuscript.
- All equations must be retyped using MathType. Ok
- How will these insulating panel perform in raining conditions? I did not see the authors
conduct any tests on the panels themselves such durability tests, permeability tests? Please see recommendation section.
- Line 276-277 need to be repharsed. Please see lines 299-308 in the revised manuscript.
- In real life, there will be some sort of ventilation. How this is taken into account in the
test program? Please see lines 204-206 in the revised manuscript.
- The conclusion section needs to be improved. Please see conclusion section
- The authors should clearly mention the limitation of their study. Please see lines 387-395 in the revised manuscript.
- A recommendation section is needed.Thanks you for your consideration. Please see recommendation section line 396 in the revised manuscript.
Thank you for your attention.
Reviewer 2 Report
Comments and Suggestions for Authors
The paper presents an interesting topic, evaluating the thermal performance of rice husk panels. However, the manuscript has some issues that must be improved, and the production process and the thermal conductivity calculation must be explained better.
The production process and the thermal conductivity calculation must be explained better.
In the abstract, the authors mentioned that the insulation panels were developed in accordance with ASTM C-177. However, this standard is applied for heat flux measurement and thermal transmission properties by means of the Guarded Hot plate. If there is no error, this must be explained.
In the abstract, the authors say that the thermal performance of rice husk-based panels was comparable with conventional insulators. It should specify at least one insulation material with similar thermal conductivity, around 0.073 W/(mK).
In line 43, it would be worth mentioning the percentage of heat that enters the space through the roof, as the authors stated that it is significant, and the paper is based on this fact.
In line 54, thermal conductivity is W/(mºC); on line 56, it is expressed as W/(mK). To be coherent, please consider using just the second unit.
In lines 57-58, the authors refer to hectares and then to quintals. It is suggested that only one unit of measure is considered for better comparison and understanding.
In line 109, correct English spelling. From “it is analyzed it potential as…” to “it is analyzed its potential as…”
In Section 2, it is not mentioned the accuracy and the measurement range of the equipment used.
If possible, section 2.1 should include images of the rice rusk panel production process, such as binder consistency, the aluminum mold used, and the final sample.
In line 182, as well as in table 1, it is stated that the temperature inside the “hotbox” reached 230ºC, is it the correct temperature? DHT22 temperature sensors do not have this range of temperature.
In Table 1, heat flux is considered 85W, the authors should explain how this value was obtained. Furthermore, the heat flux unit should be W/m2.
In line 196, the equation result should be confirmed due to inconsistencies in values.
In lines 198-199, a reference must be added.
Figure 3 should be better explained and more detailed. The location of the cardboard sheets and rice husk panels is not clear.
Identify Figures 6a and 6b.
In conclusions or in discussion, the fire resistance issue should be mentioned, why it was not considered, whether it is relevant, etc.
Author Response
May 7, 2024
Dear Editors:
I would like to submit the manuscript «materials-2999973 », entitled « Rice husk-based insulators: Manufacturing process and thermal potential assessment » by author names to be reconsidered for publicaction in Development and Characterization of Bio-Based Insulation Materials, as an special issue of Materials (ISSN 1996-1944), section Construction and Building Materials
All the changes are in red, answering to reviewer’s recommendations. The changes based on reviewer recommendations are:
- The production process and the thermal conductivity calculation must be explained better. Please see lines 188-198 in the revised manuscript.
- In the abstract, the authors mentioned that the insulation panels were developed in accordance with ASTM C-177. However, this standard is applied for heat flux measurement and thermal transmission properties by means of the Guarded Hot plate. If there is no error, this must be explained. Please see the revised abstract.
- In the abstract, the authors say that the thermal performance of rice husk-based panels was comparable with conventional insulators. It should specify at least one insulation material with similar thermal conductivity, around 0.073 W/(mK). Please see the revised abstract.
- In line 43, it would be worth mentioning the percentage of heat that enters the space through the roof, as the authors stated that it is significant, and the paper is based on this fact. Please see lines 42-49 in the revised manuscript.
- In line 54, thermal conductivity is W/(mºC); on line 56, it is expressed as W/(mK). To be coherent, please consider using just the second unit. Please see lines 58-59.
- In lines 57-58, the authors refer to hectares and then to quintals. It is suggested that only one unit of measure is considered for better comparison and understanding. This information refers at first, to rice cultivated area in hectares. Then the quintal is the unit used in Panama to refers to the quantity of resulted crop and waste.
- In line 109, correct English spelling. From “it is analyzed it potential as…” to “it is analyzed its potential as…” Please see line 111 in the new manuscript.
- In Section 2, it is not mentioned the accuracy and the measurement range of the equipment used.
- If possible, section 2.1 should include images of the rice rusk panel production process, such as binder consistency, the aluminum mold used, and the final sample. Please see figure 1 in the new manuscript.
- In line 182, as well as in table 1, it is stated that the temperature inside the “hotbox” reached 230ºC, is it the correct temperature? DHT22 temperature sensors do not have this range of temperature.
- In Table 1, heat flux is considered 85W, the authors should explain how this value was obtained. Furthermore, the heat flux unit should be W/m2. 85 W refers to heat transferred through the material. This is verified.
- In line 196, the equation result should be confirmed due to inconsistencies in values. This is verified.
- In lines 198-199, a reference must be added. Please see lines 187-199 in the new manuscript.
- Figure 3 should be better explained and more detailed. The location of the cardboard sheets and rice husk panels is not clear. Please see lines 241-247 in the new manuscript. Figure 3 now is figure 4.
- Identify Figures 6a and 6b.Please see lines 294-298 in the new manuscript.
- In conclusions or in discussion, the fire resistance issue should be mentioned, why it was not considered, whether it is relevant, etc. Please see recommendations section.
Thank you for your attention.
Round 2
Reviewer 1 Report
Comments and Suggestions for Authors
the authors significantly improved the quality of their manuscript and properly addressed my comments. i have no reservations
Author Response
May 13, 2024
Dear Editors:
I would like to submit the manuscript entitled « Rice husk-based insulators: Manufacturing process and thermal potential assessment » by author names to be considered for publication in Development and Characterization of Bio-Based Insulation Materials, as an special issue of Materials (ISSN 1996-1944), section Construction and Building Materials.
All the changes are in red, answering to reviewers’ recommendations.
Thanks you for your consideration.
Reviewer 2 Report
Comments and Suggestions for Authors
The authors made some improvements, but some points remain unclear, such as:
- In the abstract , the sentence “Square rice husk-based insulation panels were developed in accordance with ASTM C-22 177.”, must be explained as the mentioned standard does not refer to product development. Could you clarify how this standard is relevant to the development of the product?
- Avoid using quintals to indicate the quantity of harvested crops. Instead, convert the value to its equivalent in kilograms or square meters to ensure standardized units and internationally recognizable.
- The author mentioned a correction factor used for experiments exceeding the manufacturer's indicated temperature limits. Is it described in the sensor's manual? How a sensor with a maximum measurement range can measure up to 230ºC? Could you provide a detailed explanation of this correction factor, its source, and how it ensures the reliability of the calculations?
- Better explain this phrase, in lines 194-195: “To obtain the correction factor, the ambient temperature at the test location was measured and divided by the constant equivalent to that temperature”. What is the value of the constant equivalent and how did you find it?
- Figure 4 still lacks clarity regarding the location of the cardboard sheets and rice husk panels. Please improve Figure 4.
The authors could reply to reviewers in a more detailed and direct way to facilitate the understanding of the author’s point of view and arguments.
Author Response
May 13, 2024
Dear Editors:
I would like to submit the manuscript entitled « Rice husk-based insulators: Manufacturing process and thermal potential assessment » by author names to be considered for publication in Development and Characterization of Bio-Based Insulation Materials, as an special issue of Materials (ISSN 1996-1944), section Construction and Building Materials.
All the changes are in red, answering to reviewers’ recommendations. The changes based on reviewer’s recommendations are:
- In the abstract, the sentence “Square rice husk-based insulation panels were developed in accordance with ASTM C-22 177.”, must be explained as the mentioned standard does not refer to product development. Could you clarify how this standard is relevant to the development of the product?
The text change to: Square rice husk-based insultation panels were developed, considering ASTM C-177 dimensions to perform thermal conductivity coefficient tests. The ASTM C-177 is used to obtain the value of k-value (thermal conductivity coefficient) of the developed panels. In this standard, the required dimensions are: 30 cm wide, 30 cm high and 1.5 cm thickness.
- Avoid using quintals to indicate the quantity of harvested crops. Instead, convert the value to its equivalent in kilograms or square meters to ensure standardized units and internationally recognizable.
The text was changed to: About 98 040 hectares of rice crop are cultivated in Panama and the rice husk harvest reached 812 740 000 kilograms, between 2020-2021.
- The author mentioned a correction factor used for experiments exceeding the manufacturer's indicated temperature limits. Is it described in the sensor's manual? How a sensor with a maximum measurement range can measure up to 230ºC? Could you provide a detailed explanation of this correction factor, its source, and how it ensures the reliability of the calculations?
DTH22 is a basic, low-cost digital temperature and humidity sensor. This sensor uses a capacitive humidity sensor and a thermistor to measure the surrounding air and spits out a digital signal.
The manufacturer recommends the use temperature compensation, because it depends on ambient conditions. In previous work (Carvajal Flores, R.; Solís Centella, J.L.; Marín Calvo, N. Prototipo de Medida de Conductividad Térmica de Materiales Basado En La Norma ASTM C177. In Proceedings of the Congreso Internacional de Investigación e Innovación, Cortázar, Guanajuato, México; 2020; pp. 878–887) the procedure to obtain thermal conductivity coefficient (K-values) was stablished (with the “hot box” equipment), performing tests with different materials with known K-value, at the same ambient conditions. Considering this previous experience, the correction factor was calculated the ambient temperature in Panama (30 °C) by the corresponding resistance value for indoor and outdoor ambient temperature sensors (12.07). The correction factor obtained was 2.4855. The temperature provided by the sensor was automatically multiplied by this factor, thus providing the actual temperature inside the “hot box”.
- Better explain this phrase, in lines 194-195: “To obtain the correction factor, the ambient temperature at the test location was measured and divided by the constant equivalent to that temperature”. What is the value of the constant equivalent and how did you find it?
The text changes to: To obtain the correction factor, the ambient temperature at the test location was measured and divided by the constant equivalent to that temperature. Considering ambient temperature of 30°C and the corresponding resistance value for indoor and outdoor ambient temperature sensors (12.07). Divided the temperature by the resistance value results in the correction factor of 2.4855. The temperature provided by the sensor was automatically multiplied by this factor, thus providing the actual temperature inside the “hot box”. The previous procedure was stablished, performing tests with different materials with known K-value, at the same ambient conditions [43].
- Figure 4 still lacks clarity regarding the location of the cardboard sheets and rice husk panels. Please improve Figure 4.
A new view added in the figure 4 shows the location of the material under the galvanized roof: the rice husk panels were placed under the galvanized roof, then the compressed cardboard sheets were installed under the rice husk panels, as insulation system in one of the chambers (figure 4 b). Please see the new figure 4.
Thanks you for your consideration.

Round 3
Reviewer 2 Report
Comments and Suggestions for Authors
The authors have improved the manuscript. In my opinion, the paper can be accepted.
I requested a validation of the method, supported by other works, employed to measure heat transfer, a crucial aspect of the research. The authors have indeed presented a conference paper utilizing the same method, but two of the authors are also contributors to this one. I do not question the method, but seeing it applied in other works would have been relevant.